# Variation of tuberculosis prevalence across diagnostic approaches and geographical areas of Indonesia

**Alvera Noviyani[1,2], Tanawin Nopsopon[3], Krit Pongpirul** [1,3,4] *

**1** Health Development Program, Faculty of Medicine, Chulalongkorn University, Bangkok, Thailand,
**2** Faculty of Public Health, Sriwijaya University, Kota Palembang, Sumatera Selatan, Indonesia, **3** School of Global Health, Faculty of Medicine, Chulalongkorn University, Bangkok, Thailand, **4** Department of International Health, Johns Hopkins Bloomberg School of Public Health, Baltimore, Maryland, United States of America

\* doctorkrit@gmail.com

**Data Availability Statement:** The data in this study has legal or ethical restriction as there is a written statement between the researcher and the Indonesian Ministry of Health on sharing a de-identified data set because the data contains

## Abstract

### Background

Tuberculosis (TB) has contributed to a significant disease burden and economic loss worldwide. Given no gold standard for diagnosis, early identification of TB infection has been challenging. This study aimed to comparatively investigate the prevalence of TB across diagnostic approaches (sputum AFB, sputum culture, sputum genetic test, and chest x-ray) and geographical areas of Indonesia.

### Methods

Participant demographic variables and TB screening test results were obtained from the Tuberculosis Unit, Health Research and Development Agency, Ministry of Health (HRDA-MoH). The prevalence of pulmonary TB in populations aged 15 years and over was calculated using TB cases as a numerator and populations aged 15 years and over as a denominator. Variations across geographical areas and diagnostic approaches were expressed as prevalence and 95% confidence interval (CI).

### Results

A total of 67,944 records were reviewed. Based on bacteriological evidence, the prevalence of TB per 100,000 in Indonesia was 759 (95% CI: 589.7–960.8) with variations across areas: 913 (95% CI 696.7–1,176.7; Sumatra), 593 (95% CI 447.2–770.6; Java-Bali), and 842 (95% CI 634.7–1,091.8; other islands). Also, the prevalence of TB varied across diagnostic approaches: 256.5 (sputum AFB), 545 (sputum culture), 752.2 (chest x-ray), and 894.9 (sputum genetic test). Based on sputum AFB, the TB prevalence varied from 216.6 (95% CI 146.5–286.8; Java-Bali), 259.9 (95% CI 184.2–335.6; other islands) to 307.4 (95% CI 208.3–406.5; Sumatra). Based on sputum culture, the TB prevalence ranged from 487.9 (95% CI 433.6–548.6; Java-Bali), 635.9 (95% CI 564.9–715.1; Sumatra), to 2,129.8 (95% CI 1,664.0–2,735.6; other islands). Based on chest x-ray, the TB prevalence varied from

potentially sensitive information, and the data are owned by a third-party organization (Indonesian Ministry of Health). Requests for data must submit formally to the Head of the Health Research and Development Agency, the Indonesian Ministry of Health. The data can be requested or accessed through Health Research and Development Agency, Indonesian Ministry of Health (layanan. data@litbang.kemkes.go.id).

**Funding:** AN received the ASEAN Scholarship 2019 from Chulalongkorn University.

**Competing interests:** The authors have declared that no competing interests exist.

152.1 (95% CI 147.9–156.3; Java-Bali), 159.2 (95% CI 154.1–164.3; Sumatra), to 864 (95% CI 809–921.4; other islands). Based on sputum genetic test, the TB prevalence ranged from 838.7 (95% CI 748.4–900.8; Java-Bali), 875 (95% CI 775.4–934.2; Sumatra), to 941.2 (95% CI 663.6–992.3; other islands).

## Conclusions

The variation of TB prevalence across geographical regions could be confounded by the diagnostic approaches.

## Trial registration

This study was approved by the Institutional Review Board of Chulalongkorn University (IRB No. 684/63).

## Introduction

Tuberculosis (TB) was one of the top diseases which caused numerous death worldwide [1]. TB was a widespread infectious disease caused by the bacterium *Mycobacterium tuberculosis* [2, 3], which typically affected the lungs (pulmonary TB) but could also affect other organs (extra-pulmonary TB) [3–5]. TB has continued to remain a hot issue and a major health problem, especially in developing countries [6, 7] which accounted for approximately 95% of TB-related deaths [8]. In 2019, 10.0 million people fell ill with TB, including an estimated 1.2 million children, with approximately 1.4 million TB deaths, including 208 thousand people with HIV/AIDS, which mostly induced by poverty and vulnerability of the community [1, 9]. While the End TB Strategy has been implemented worldwide, declining trends of TB infection and deaths were relatively slow and fell short of the 2020 goal [1]. There were many challenges in efforts to prevent and control global TB. Some of the real challenges faced were significant gaps in funding, lack of access to diagnostic and treatment services, limitations in resources, the emergence, and transmission of multi-drug resistant TB strains in endemic countries, particularly Asian countries [10].

An estimated around 3.6 million people with TB were missed diagnosis by health systems every year, thus did not receive adequate care [11]. Missed diagnosis, particularly on early-stage TB patients who developed mild or no symptoms, led to high numbers of TB infected people arrived at clinics with advanced disease or multi-drug resistant TB (MDR-TB), which was difficult to treat and more likely to cause death [12–14]. In 2019, the largest number of new TB cases occurred in the South-East Asia region, with 44% of global new cases, followed by the African region with 25% of global new cases, and the Western Pacific with 18%. While TB occurred in every part of the world, eight countries comprised two-thirds of global TB incidence which Indonesia ranks second for the incidence of TB in the world after India [1, 9]. The high incidence of TB in Indonesia indicated a high priority for TB prevention and control [15].

Based on the Center for Disease Control and Prevention 2016, Tuberculosis could be found through various testing and diagnosis [16, 17]. TB mycobacteria in the body could be identified with two types of the test: TB skin test (TST) and TB blood tests, while the results only presented that a person has been infected with TB. Other tests, such as a chest x-ray and a sample of sputum, were needed to differentiate whether the person had latent or active TB infection [16].

One of the key strategies for tackling TB was an early diagnosis. Unfortunately, sputum testing was only 50% accurate and frequently missed the disease in its early stages [11]. Systematic screening strategy to ensure early and correct diagnosis for all people with TB for instance chest X-ray (CXR) was one of the primary tools for TB screening [18]. There were various methods to diagnose TB disease, although every method had an error and could lead to a missed diagnosis. In detecting tuberculosis, inadequate access to advanced diagnostic tests contributed to suboptimal detection performance [19]. Delay in diagnosis and misdiagnosis led to increased morbidity and mortality in patients and allowed continued TB transmission [19, 20]. In many countries with a high burden of TB, there was geographically substantial heterogeneity in the burden of TB. Besides, decisions about TB funding and policy were usually very decentralized [21]. Thus, data could refer geographically to priority areas as needed.

This research aimed to comparatively investigate the prevalence of TB across diagnostic approaches, including sputum acid-fast bacilli (AFB), sputum culture, sputum genetic test, and chest x-ray, and geographical areas in Indonesia. The area with a high prevalence should be put on the priority list in the TB prevention program.

## Materials and methods

### Study design

This study analyzed the screening results of Indonesian people using various diagnostic approaches (sputum AFB, sputum culture, sputum genetic test, and chest x-ray) in the data set obtained from the Health Research and Development Agency of the Tuberculosis Unit, Indonesian Ministry of Health (HRDA-MoH). The primary study was carried out using a cross-sectional design which was carried out utilizing interviews, measurements, and chest X-ray examination, and sputum examination for all samples. The sample selection was stratified multi-stage cluster sampling in the public. Training and pilot tests, as well as cleaning, validation, and data analysis, had been completed.

### Study setting

The data set represented Indonesia in all of the 34 provinces, which were classified into 3 large islands in Indonesia, Sumatra island, Java-Bali island, and other islands (other islands with a relatively small population which are mostly in the eastern part of Indonesia). Characteristics of the regions are presented in Table 1. Indonesia as the largest archipelagic country in the world had two-thirds of its territory in the form of Indonesian oceans, namely 6.32 million square kilometers, 17,504 islands, and was one of the countries that had the longest coastline in the world after Canada. So, it was necessary to divide the time into 3 zones time in Indonesia, WIB (Western Indonesian Time), WITA (Central Indonesian Time), and WIT (Eastern Indonesian Time). There could be a time difference of up to 8 hours from one island to another. Besides, geographically, Indonesia was located between two continents (Asian Continent and Australian Continent) and two oceans (the Indian Ocean and the Pacific Ocean) which were the most dynamic regions, both economically and politically. This strategic geographical location made Indonesia both superior and highly dependent on the marine [22].

### Study population

Individuals aged 15 years and over who had stayed in the selected cluster for at least one month. Individuals living in military barracks, diplomatic mission houses, hospitals, hotels, dormitories, temporary residences were excluded.

**Table 1. Characteristics of the regions in general.**

| Characteristics | Sumatra | Java-Bali | Others | References |
|---|---|---|---|---|
| **Total population** | 58.56 million | 151.59 million | 60.05 million | [32] |
| **Gender** | | | | |
| **Male** | 29,711,506 | 76,385,599 | 30,564,794 | [32] |
| **Female** | 28,845,705 | 75,205,663 | 29,490,650 | [32] |
| **Migration rates** | | | | |
| **Moving in** | 7,889,399 | 15,236,040 | 6,656,315 | [29] |
| **Moving out** | 6,669,677 | 18,329,895 | 4,585,403 | [29] |
| **Noncommunicable diseases** | | | | |
| **Hypertention** | 136,569 | 384,746 | 136,886 | [29] |
| **Diabetes mellitus** | 149,142 | 415,232 | 149,412 | [29] |
| **Nutritional Status by BMI (>18 years)** | | | | |
| **Skinny** | 10466.53 | 34791.56 | 12926.95 | [29] |
| **Overweight** | 17615.35 | 49730.9 | 17312.02 | [29] |
| **Obesity** | 28103.87 | 82507.31 | 25844.29 | [29] |
| **Infectious diseases** | | | | |
| **HIV** | 6,217 | 29,689 | 13,818 | [33] |

## Statistical analysis

Validated and merged data were exported from a Microsoft Access-based database, transformed with SPSS version 22 and Stata software version 15.0. Categorical data were presented with counts and percentages. The prevalence of pulmonary TB in populations aged 15 years and over was calculated using TB cases as a numerator and populations aged 15 years and over as a denominator which is consistent with the WHO's global tuberculosis guidance [23]. Variations across geographical areas and diagnostic approaches were expressed as prevalence with a 95% confidence interval (CI) using Clopper-Pearson Exact method.

## Diagnostics tests

**Chest x-ray.** After data collection was complete, the chest X-ray image was sent to the central team and submitted to the central reader. Three radiologists, without knowing the results of the field screening (blinded), interpreted the results of the chest X-ray. The classification of the results of the central reading was divided into (1) normal, (2) abnormal lung parenchyma or pleura which consists of TB and non-TB features. The picture of TB consists of seven abnormalities, namely infiltrates nodule consolidation, cavitary lesions, fibrosis, calcification, pleural effusion, and pleural thickening, (3) others. The results of the field chest X-ray reading were used to decide which participants to take sputum from. The results of the central reading were used to decide the case definition.

**Sputum AFB.** Diagnosis of TB by identifying acid-fast bacilli (AFB) in sputum that is not concentrated (direct smear) with Ziehl-Neelsen (ZN).

**Sputum culture.** TB diagnosis uses selective media for the cultivation and isolation of Mycobacterium species. Lowenstein-Jensen (LJ) medium is most widely used for tuberculosis culture in Indonesia.

**Sputum genetic test.** The Xpert MTB/RIF assay is an automated, cartridge-based nucleic acid amplification test that uses the multi-disease GeneXpert platform.

## Ethics committee approval

This study was approved by the Institutional Review Board of Chulalongkorn University (COA No. 1316/2020).

### Patient and public involvement

Patients and the public have not been directly involved in the design, conduct, or reporting of this study. However, we believe that our study would provide additional information for patients who lived in or would travel to relatively high TB incidence areas and for the public to have a more accessible real-world experience.

## Results

### Demographic characteristics of the participants

Indonesia consisted of 17,491 islands in 34 provinces, including the Special Capital Region Jakarta. The province consisted of districts (district) and municipality (city); the city was further divided into districts, which were then divided into administrative villages [24], the following was the presentation of population distribution that was eligible as participants according to demographic characteristics (Table 2). There were six groups of participants according to the age years (15–24 years old, 25–34 years old, 35–44 years old, 45–54 years old, 55–64 years old, and 65 years old and over). The region classification of the study was divided into three big islands in Indonesia: Sumatra island, Java-Bali island, and other islands (other islands with a relatively small population which are mostly in the eastern part of Indonesia) with urban and rural as an area classification.

### Tuberculosis prevalence

The prevalence of tuberculosis was 759.1 (95% CI: 589.7–960.8) per 100,000 population aged 15 years and over with an increasing trend of TB prevalence with aging. The TB prevalence

**Table 2. Distribution of eligible populations and participants according to demographic characteristics.**

| Characteristics | Eligible | Participants | Non-participants |
|---|---|---|---|
| | n (%) | n (%) | n (%) |
| Total | 76,576 | 67,944 | **8,632** |
| Age group (years)* | | | |
| 15–24 | 16,982 (22.2) | 14,505 (21.3) | **2,477 (28.7)** |
| 25–34 | 17,760 (23.2) | 15,192 (22.4) | **2,568 (29.7)** |
| 35–44 | 16,107 (21.0) | 14,386 (21.2) | **1,721 (19.9)** |
| 45–54 | 12,677 (16.6) | 11,643 (17.1) | **1,034 (12.0)** |
| 55–64 | 7,410 (9.7) | 6,870 (10.1) | **540 (6.3)** |
| 65 or more | 5,640 (7.4) | 5,348 (7.9) | **292 (3.4)** |
| Gender | | | |
| Male | 36,759 (48.0) | 31,632 (46.6) | **5,127 (59.4)** |
| Female | 39,817 (52.0) | 36,312 (53.4) | **3,505 (40.6)** |
| Area Classification | | | |
| Urban | 37,865 (49.4) | 31,871 (46.9) | **5,994 (69.4)** |
| Rural | 38,711 (50.6) | 36,073 (53.1) | **2,638 (30.6)** |
| Region* | | | |
| Sumatra | 22,700 (29.6) | 19,739 (29.1) | **2,961 (34.3)** |
| Java-Bali | 31,049 (40.5) | 28,150 (41.4) | **2,899 (33.6)** |
| Other | 22,827 (29.8) | 20,055 (29.5) | **2,772 (32.1)** |

Data were presented in counts and percentages.

*Total percentage was not equal to 100% due to rounding.

**Table 3. Estimated TB prevalence per 100,000 population aged 15 years and over according to demographic characteristics and diagnostic approach.**

| Characteristics | Observed | | Chest x-ray | | Sputum AFB | | Sputum Culture | | Sputum Genetic Test | |
|---|---|---|---|---|---|---|---|---|---|---|
| | Prevalence | 95% CI | Prevalence | 95% CI | Prevalence | 95% CI | Prevalence | 95% CI | Prevalence | 95% CI |
| **Total** | 759.1 | 589.7–960.8 | 725.2 | 718.1–732.2 | 256.5 | 210.1–302.9 | 545.0 | 509.5–583.0 | 894.9 | 848.7–928.2 |
| **Age group (years)** | | | | | | | | | | |
| **15–24** | 360.8 | 254.3–494.7 | 783.8 | 741.2–828.7 | 137.5 | 77.3–197.8 | 414.5 | 330.7–518.3 | 851.8 | 659.6–944.6 |
| **25–34** | 753.4 | 561.8–995.0 | 109.1 | 104.3–114.2 | 239.9 | 155.5–324.4 | 561.4 | 477.0–659.7 | 893.6 | 766.1–955.6 |
| **35–44** | 713.8 | 527.4–941.0 | 142.9 | 137.3–148.7 | 265.1 | 170.7–359.4 | 584.7 | 503.8–677.5 | 933.3 | 809.7–978.7 |
| **45–54** | 835.5 | 608.9–1,108.3 | 205.7 | 198.5–213.2 | 271.5 | 166.3–376.7 | 479.7 | 408.9–561.8 | 891.3 | 761.4–954.6 |
| **55–64** | 1,029.5 | 734.1–1,398.5 | 284.7 | 274.2–295.5 | 318.6 | 174.1–463.1 | 529.6 | 446.5–627.2 | 909.1 | 748.6–971.1 |
| **65 or more** | 1,581.7 | 1,122.7–2,153.7 | 373.7 | 360.9–386.8 | 527.6 | 292.0–763.2 | 678.9 | 583.6–788.6 | 875.0 | 729.5–947.8 |
| **Gender** | | | | | | | | | | |
| **Male** | 1,082.7 | 872.8–1,337.3 | 200.8 | 196.4–205.3 | 392.5 | 314.5–470.5 | 607.9 | 558.6–661.2 | 896.9 | 839.9–935.2 |
| **Female** | 460.6 | 353.6–590.8 | 133.5 | 130.1–137.1 | 131.0 | 87.6–174.4 | 463.0 | 414.2–517.3 | 890.4 | 794.4–944.7 |
| **Area Classification** | | | | | | | | | | |
| **Urban** | 845.8 | 678.2–1,047.7 | 150.1 | 146.2–154.0 | 282.2 | 219.6–344.7 | 736.6 | 673.8–804.7 | 912.0 | 847.3–950.8 |
| **Rural** | 674.2 | 511.9–873.6 | 177.9 | 174.0–181.9 | 231.4 | 163.3–299.5 | 408.8 | 369.0–452.7 | 876.1 | 800.7–925.6 |
| **Region** | | | | | | | | | | |
| **Sumatra** | 913.1 | 696.7–1,176.7 | 159.2 | 154.1–164.3 | 307.4 | 208.3–406.5 | 635.9 | 564.9–715.1 | 875.0 | 775.4–934.2 |
| **Java-Bali** | 593.1 | 447.2–770.6 | 152.1 | 147.9–156.3 | 216.6 | 146.5–286.8 | 487.9 | 433.6–548.6 | 838.7 | 748.4–900.8 |
| **Other** | 842.1 | 634.7–1,091.8 | 864.0 | 809.0–921.4 | 259.9 | 184.2–335.6 | 2,129.8 | 1,664.0–2,735.6 | 941.2 | 663.6–992.3 |

Data were presented in counts and percentages.

AFB, Acid-fast bacilli; CI, confidence interval; TB, tuberculosis.

was higher in the 65+ age group than other groups with 1,581.7 (95% CI: 1,122.7–2,153.7) per 100,000 population (Table 3). Tuberculosis prevalence in Indonesia was higher in urban areas than in rural areas, and the Sumatra region had the highest TB prevalence with 913.1 (95% CI: 696.7–1,176.7) per 100,000 population.

### Diagnostic approach variation

**Chest x-ray.** Among participants who were screened positive, 43,6% had no symptoms of cough 14 days or more or blood cough (Table 4). There were 11,202 participants with abnormal Chest X-ray results from 15,446 total participants. Tuberculosis prevalence according to Chest X-ray examination was 725.2 (95% CI: 718.1–732.2) per 100,000 population. The prevalence of TB with positive Chest x-ray was higher at the age 15–24 years with 783.8 (95% CI: 741.2–828.7) per 100,000 population. The prevalence across the demographic region of chest

**Table 4. Distribution of screened participants.**

| Coughing ≥14 days or coughing up blood | Chest x-ray | n | % |
|---|---|---|---|
| **Yes** | Normal | 3,844 | 24.9 |
| **Yes** | Abnormal | 4,459 | 28.9 |
| **Yes** | No chest x-ray | 249 | 1.6 |
| **No** | Abnormal | 6,743 | 43.6 |
| **No** | Normal or No chest x-ray | 151 | 1.0 |
| **Total** | | 15,446 | 100.0 |

Data were presented in counts and percentages.

**Table 5. Compliance with sputum culture examination results (spot and morning sputum).**

| Culture | | Morning sputum | | | | | |
|---|---|---|---|---|---|---|---|
| | | Negative | MTB | NTM | Contamination | N/A | Total |
| Spot sputum | Negative | 3,772 | 41 | 104 | 87 | 170 | 4,174 |
| | MTB | 39 | 89 | 0 | 3 | 2 | 133 |
| | NTM | 59 | 0 | 14 | 5 | 1 | 79 |
| | Contamination | 34 | 2 | 1 | 8 | 2 | 47 |
| | N/A | 85 | 1 | 3 | 3 | 13 | 105 |
| | Total | 3,989 | 133 | 122 | 106 | 188 | 4,538 |

MTB, *Mycobacterium tuberculosis*; N/A, not available; NTM, non-tuberculosis mycobacterium.

x-ray examination results in Indonesia was higher in males with 200.8 (95% CI: 196.4–205.3) per 100,000 population than the female with 133.5 (95% CI: 130.1–137.1) per 100,000 population. The prevalence was higher in the other regions in Indonesia with 864 (95% CI 809.0–921.4) per 100,000 population than in Sumatra and Java-Bali and higher in rural areas.

**Sputum AFB.** Tuberculosis prevalence according to sputum AFB examination was 256.5 (95% CI: 210.1–302.9) per 100,000 population. The highest TB prevalence is at 65 years and over the group, while the lowest prevalence is at 15–24 years group with 527.6 (95% CI: 292.0–763.2) and 137.5 (95% CI:77.3–197.8) per 100,000 population, respectively. The prevalence of TB with a positive sputum AFB was higher in the male than the female with 392.5 (95% CI: 314.5–470.5) compared with 131.0 (95% CI: 87.6–174.4) per 100,000 population, respectively. The prevalence increased with age, the highest prevalence was 527.6 (95% CI: 292.0–763.2) per 100,000 population in the 65-year-and-over group. The finding showed that urban areas had a higher TB prevalence than rural areas. Referring to the division of regions, Sumatra island had the highest prevalence estimate by sputum AFB examination with a prevalence of 307.4 (95% CI: 208.3–406.5) per 100,000 population than in other regions.

**Sputum culture.** Sputum culture examination was taken from spot sputum and morning sputum of participants, of 106 participants with contaminated sputum culture results, 95 (89.6%) had spot sputum results and from 47 participants with contaminated spot sputum culture, 37 (78.7%) had results on sputum morning. Among negative culture results, there were 41 with MTB (*Mycobacterium tuberculosis*) positive and 104 with NTM (Non-Tuberculosis Mycobacterium) from morning culture and there were 39 with a positive spot culture of MTB (*Mycobacterium tuberculosis*), and 59 with NTM (Non-Tuberculosis Mycobacterium) in negative culture results (Table 5). Tuberculosis prevalence according to culture examination was 545.0 (95% CI: 509.5–583.0) per 100,000 population while culture-negative tuberculosis prevalence was 945.5 (95% CI: 941.7–949.0) per 100,000 population.

According to the demographic region, the prevalence of TB with positive culture was higher in the other regions in Indonesia with 2,129.8 (95% CI: 1,664.0–2,735.6) per 100,000 population than Sumatra and Java-Bali region. The prevalence was higher in male than female participants, with 607.9 (95% CI: 558.6–661.2) and 463.0 (95% CI: 414.2–517.3) per 100,000 population, respectively. The prevalence of TB with culture examination was higher in urban areas than in rural areas. The prevalence of TB with positive culture examination was higher at the age of 65 years and over with 678.9 (95% CI: 583.6–788.6) per 100,000 population.

**Sputum genetic test.** Sputum genetic test result indicated 213 M. tuberculosis (MTB) detected, which 184 MTB were not resistant to rifampicin (rifampicin susceptible), 19 MTB were resistant to rifampicin, and 10 MTB were indeterminate (Table 6). TB prevalence according to Xpert TB examination was 894.9 (95% CI: 948.7–928.2) while Xpert TB-negative

**Table 6. The results of the sputum genetic test.**

| Examination results | Xpert MTB |
|---|---|
| **Positive** | 213 |
| RIF Sensitive | 184 |
| RIF Resistance | 19 |
| RIF Indeterminate | 10 |
| **Negative** | 407 |
| **Not detected** | 22 |
| **Error/invalid** | 4 |
| **Total** | 646 |

MTB, *Mycobacterium tuberculosis*; RIF, rifampicin.

tuberculosis prevalence was 924.0 (95% CI: 614.0–1,367.0) per 100,000 population. The prevalence of TB with positive sputum genetic test was higher in the other regions of Indonesia following by Sumatra and Java-Bali region with 941.2 (95% CI: 663.6–992.3), 875.0 (95% CI: 775.4–934.2), and 838.7 (95%CI: 748.4–900.8) per 100,000 population respectively. The prevalence of TB with positive sputum genetic test examination was higher at the age of 35–44 years with 933.3 (95% CI: 809.7–978.7) per 100,000 population. Prevalence in males and females was slightly different and higher in urban areas.

## Discussion

The prevalence of TB in the Indonesian population aged 15 years and over was 759 per 100,000; the highest TB prevalence was in the old age group (55 years and over); with a significant variation of TB prevalence across geographical areas and diagnostic approaches. The prevalence of TB was higher in the male group (prevalence of positive TB sputum AFB 393 per 100,000 and TB with bacteriological confirmation 1,082 per 100,000) than female (prevalence of positive TB sputum AFB 131 per 100,000 and TB with bacteriological confirmation 461 per 100,000). This result is consistent with other countries that show a higher prevalence in males than females [25–28]. This is probably because a male is more exposed to TB risk factors such as smoking. Indonesia's basic health research data shows that the smoking behavior of the Indonesian population aged 15 years and over, amounted to 33.8% in 2018. Men who smoke every day or smoke occasionally are 47.3%, while women are only 1, 2% [29]. Tuberculosis prevalence in Indonesia was higher in urban areas than rural areas, and the Sumatra region had the highest prevalence of TB with 913 (95% CI: 696.7–1,176.7). This could occur because there was more access to health services in the islands of Sumatra and Java but was slightly different for island areas in central and eastern Indonesia. This happened because the two islands were still underdeveloped areas.

The survey participation rate (88.7%) indicates that the survey results can be generalized. These results are not much different from other countries, such as Cambodia 89.2% [27], Myanmar 92.6% [28], and the Philippines 76% [26]. The participation was higher amongst women (53.4%) than men (46.6%) and amongst those in rural areas (53.1%) than in the urban areas (46.9%). According to the tuberculosis prevalence of the Philippines which has similar island patterns with Indonesia, the Philippines has similar results as a participation rate was higher amongst women and rural areas, and TB prevalence was higher in the old age group [26].

The prevalence of TB with positive chest x-ray was higher at the age of 15–24 years, higher in males than females, and higher in rural areas. The proportion of participants aged 65 years

and over who underwent chest x-ray was lower than younger ages. Most of the reason was that they were unable to visit the study sites because of illness, disability, or unable to walk. The proportion of female participants who underwent chest x-rays was lower than the male because some pregnant women did not have chest x-rays. The TB prevalence was higher in the other regions in Indonesia than in Sumatra and Java-Bali. Apart from the possibility that the participants in these areas were more frequently exposed to TB germs and TB risk factors, this might be because the area was still underdeveloped so that lack of education could be one of the determining factors.

As symptom-based case finding was not optimal, the passive TB case detection that has been carried out so far might also have contributed to the delay in TB diagnosis and treatment. The limitations of case finding or symptom detection tools and access to health services needed to be improved. Total 26.1% of participants who went to health personnel showed positive screening symptoms. TB cases obtained from positive screening were 245 (57.5%). Among participants who were screened positive, 44% had no symptoms of cough 14 days or more or coughing up blood. Tuberculosis prevalence according to chest x-ray examination is 725.2 per 100,000 population. The use of chest x-ray increased the number of asymptomatic TB cases by 181 (73.9% of symptomatic cases). If the chest x-ray were not used in screening, this study would have lost 42.5% of cases. This meant that the use of chest x-rays could improve case findings. Intensive case finding needed to be supported by all levels of society, including communities and community organizations that cared about TB.

The prevalence of TB with a positive smear was higher in males than females. The prevalence increased with age, urban areas were higher than rural areas, and Sumatra was higher than in other regions. The higher prevalence in urban areas was different from other studies in China which indicated an association of TB prevalence and rural residence [30]. Generally, in urban areas, the participation rate was relatively low although the TB prevalence in urban areas was not lower than in rural areas, the low participation rate could estimate the interval from urban prevalence to be wide. The high prevalence of TB in the island of Sumatra could be caused by the density and many cities and districts on the island, which resulted in more people being directly exposed to TB germs and TB sufferers. This could also be caused by the limited geographical location of other island areas, making it difficult to send samples. The accuracy of TB diagnosis was reduced when only a microscopic examination is used. Contamination in morning sputum (3%) was higher than spot sputum (1.1%). This difference was suspected because when removing the sputum spots, the participants were accompanied by laboratory staff, while the issuance of morning sputum in their respective homes was unaccompanied. The results showed that most of the pulmonary TB cases were associated with sputum specimens which are like cases in Saudi Arabia [31].

The limitations of this study are the lack of genomic sequencing and the availability of negative culture results. Other additional accurate examinations were needed to enhance the quality of the microscopic examination. Sputum genetic test examination was proven to reduce false-positive results. Since this study was conducted in a community, further research to see this trend in routine health services is needed to update the TB diagnostic algorithm. Tuberculosis prevalence according to sputum AFB examination, culture examination, and sputum genetic test examination was 256.5, 545, and 894.9 per 100,000 population, respectively.

The prevalence of TB with positive culture was higher in males than females who lived in urban areas. The prevalence was higher in the other regions in Indonesia than in Sumatra and Java-Bali because the central and eastern regions of Indonesia were still remote and underdeveloped areas, so there might be many TB risk factors and the lack of availability of access to health services that affect the handling and prevention of TB in these areas. The high number of non-TB positive smears showed that a more accurate but simple method of diagnosis such

as a sputum genetic test was needed to reduce false-positive rates and unnecessary treatment. The prevalence of TB with positive sputum genetic test was higher in males than the female with a slight difference, and higher in urban areas. The prevalence was higher in the other regions in Indonesia than in Sumatra and Java-Bali. Epidemiological studies show that the prevalence of TB is decreasing by about 4% per year, indicating that incidence decreased by about 2.4% per year. Thus, several factors are likely to contribute to the reduction in TB incidence in Indonesia; for example, rapid improvements in the economy and other related social and economic factors.

## Conclusion

TB prevalence varies across geographical areas of Indonesia and could be confounded by the diagnostic approaches. The prevalence of pulmonary TB in Indonesia with bacteriological confirmation was 759.0 per 100,000 population aged 15 years and over with higher prevalence in urban areas and the Sumatra region had the highest prevalence. TB prevalence was 725.2 with chest x-ray, 256.5 with sputum AFB, 545.0 with sputum culture, and 894.9 per 100,000 population with sputum genetic test examination. Indonesia is a vast country. It is undeniable that the spread of TB varies in each geographic area. Adding potential human resources need to be considered, as educational background can be part of understanding tuberculosis, and limited transportation facilities caused by the location between islands which are quite far from each other. Furthermore, a limited number of health workers, health facilities, and medical equipment could be a significant problem for Indonesia.

## Acknowledgments

The authors would like to thank the Ministry of Health of the Republic of Indonesia for providing and allowing the authors to use the data. Thanks to reviewers for constructive and positive comments.

## Author Contributions

**Conceptualization:** Alvera Noviyani, Krit Pongpirul.

**Data curation:** Alvera Noviyani, Tanawin Nopsopon, Krit Pongpirul.

**Formal analysis:** Alvera Noviyani, Tanawin Nopsopon, Krit Pongpirul.

**Investigation:** Alvera Noviyani, Tanawin Nopsopon, Krit Pongpirul.

**Methodology:** Tanawin Nopsopon, Krit Pongpirul.

**Project administration:** Alvera Noviyani, Krit Pongpirul.

**Resources:** Alvera Noviyani, Krit Pongpirul.

**Software:** Alvera Noviyani, Tanawin Nopsopon, Krit Pongpirul.

**Supervision:** Krit Pongpirul.

**Validation:** Alvera Noviyani, Tanawin Nopsopon, Krit Pongpirul.

**Visualization:** Tanawin Nopsopon.

**Writing – original draft:** Alvera Noviyani, Krit Pongpirul.

**Writing – review & editing:** Alvera Noviyani, Tanawin Nopsopon, Krit Pongpirul.

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
