## [Decision Letter · Decision Letter 0]

30 Jul 2021

PONE-D-21-15288

Variation of tuberculosis prevalence across diagnostic approaches and geographical areas of Indonesia.

PLOS ONE

Dear Dr. Pongpirul,

Thank you for submitting your manuscript to PLOS ONE. After careful consideration, we feel that it has merit but does not fully meet PLOS ONE’s publication criteria as it currently stands. Therefore, we invite you to submit a revised version of the manuscript that addresses the points raised during the review process.

Please submit your revised manuscript. If you will need significantly more time to complete your revisions, please reply to this message or contact the journal office at plosone@plos.org. Please include the following items when submitting your revised manuscript:

We look forward to receiving your revised manuscript.

Kind regards,

Frederick Quinn

Academic Editor

PLOS ONE

3. Thank you for stating the following in the Acknowledgments/ Funding Section of your manuscript:

“ASEAN Scholarships 2019 by Chulalongkorn University.”

“AN received the ASEAN Scholarship 2019 from Chulalongkorn University.”

Please include your amended statements within your cover letter; we will change the online submission form on your behalf."

Additional Editor Comments (if provided):

Reviewers' comments:

Reviewer's Responses to Questions

**Comments to the Author**

1. Is the manuscript technically sound, and do the data support the conclusions?

Reviewer #1: Yes

Reviewer #2: Yes

2. Has the statistical analysis been performed appropriately and rigorously? 

Reviewer #1: Yes

Reviewer #2: Yes

3. Have the authors made all data underlying the findings in their manuscript fully available?

Reviewer #1: Yes

Reviewer #2: Yes

4. Is the manuscript presented in an intelligible fashion and written in standard English?

Reviewer #1: Yes

Reviewer #2: Yes

5. Review Comments to the Author

Reviewer #1: The authors analyzed 67,944 records and report the prevalence of TB in different areas of Indonesia comparing sputum AFB, sputum culture, sputum genetic test, and chest radiograph. The authors use only the prevalence and the confidence interval to compare the different groups. The study reports the classification of three large islands in Indonesia: the island of Sumatra, the island of Java-Bali, and other islands (eastern part of Indonesia) with a classification of urban and rural areas.

Although the title is attractive, they do not conduct any geographic study.

The authors did not discuss the prevalence of TB in other developing countries, which would contribute to the conceptualization of TB at the global level.

Although they conclude that diagnostic approaches confuse tuberculosis, they do not mention full genomic sequencing as an alternative.

Then it would be convenient to modify the discussion by including data from other countries.

Reviewer #2: This is an interesting study represents an analysis of Variation of TB prevalence across diagnostic approaches and 5 geographical areas of Indonesia

This article could be published in the Journal PLOS ONE after a major revision. I have some comments on it:

Introduction: In line 53 “MTb” name must be in italics; review of the rest of the document.

Methodology: The methodology does not describe the diagnostic tests used and the quality standards used to validate the results. Explain more about the methods used.

Results: The authors must show the risk factors of the patients (HIV, DM, malnutrition, immigration, etc.) and their association with the results in the diagnostic tests, in a table.

• What is the proportion of negative bk and positive culture in the different geographical areas? And how much is the culture, contributing to the timely diagnosis in these cases?.

• How do the authors explain that the prevalence of the positive culture is higher in the eastern region of Indonesia, compared to the results of the Chest X-ray and sputum AFB methods where the positive results are higher for others regions?.

• The authors could include data on the prevalence of migration-related TB cases, do you have any data on tuberculosis in migrants that you can include?.

• In your study you didn’t differentiate between active and latent TB, or the observed results include both?.

The discussion part is very short, need to discuss more issues of “person’ place, and time”.

In the conclusion, could the authors explain from their point of view, some reasons for the spread of TB in the different geographical areas of Indonesia and their strategies or suggestions to improve early detection?.

• The article has some mistakes in English language and have to be corrected.

6. PLOS authors have the option to publish the peer review history of their article (what does this mean?). If published, this will include your full peer review and any attached files.

Reviewer #1: No

Reviewer #2: **Yes: **Armando Martínez-Guarneros

---

## [Author Response · Author response to Decision Letter 0]

22 Sep 2021

Dear Editor,

We thank you and the reviewers for the comments and suggestions. Please find our point-by-point responses below.

Editor: Thank you for submitting your manuscript to PLOS ONE. After careful consideration, we feel that it has merit but does not fully meet PLOS ONE’s publication criteria as it currently stands. 1. Please ensure that your manuscript meets PLOS ONE's style requirements, including those for file naming.

Response: Thank you for your time and consideration. The manuscript was revised to meet PLOS ONE’s publication criteria as well as the style requirements.

Editor: 2. We note that the grant information you provided in the ‘Funding Information’ and ‘Financial Disclosure’ sections do not match. When you resubmit, please ensure that you provide the correct grant numbers for the awards you received for your study in the ‘Funding Information’ section.

Response: Thank you very much. We have updated the grant information both in ‘Funding Information’ and ‘Financial Disclosure’ sections.

Editor: 3. Thank you for stating the following in the Acknowledgments/ Funding Section of your manuscript: “ASEAN Scholarships 2019 by Chulalongkorn University.” We note that you have provided funding information that is not currently declared in your Funding Statement. However, funding information should not appear in the Acknowledgments section or other areas of your manuscript. We will only publish funding information present in the Funding Statement section of the online submission form. Please remove any funding-related text from the manuscript and let us know how you would like to update your Funding Statement. Currently, your Funding Statement reads as follows: “AN received the ASEAN Scholarship 2019 from Chulalongkorn University.” 

Response: Thank you so much for your direction. We are sorry for the confusion and thank you very much for pointing that out. We removed the funding information from the Acknowledgment Section and other areas of the manuscript accordingly. The Funding Statement is correct and can be presented as it currently reads.

Editor: 4. We note that you have indicated that data from this study are available upon request. PLOS only allows data to be available upon request if there are legal or ethical restrictions on sharing data publicly. In your revised cover letter, please address the following prompts:

Response: The data in this study has legal or ethical restriction as there is a written statement between the researcher and the Indonesian Ministry of Health on sharing a de-identified data set because the data contains potentially sensitive information, and the data are owned by a third-party organization (Indonesian Ministry of Health). Requests for data must submit formally to the Head of the Health Research and Development Agency, the Indonesian Ministry of Health. The data can be requested or accessed through Health Research and Development Agency, Indonesian Ministry of Health (layanan.data@litbang.kemkes.go.id).

Editor: 5. Your ethics statement should only appear in the Methods section of your manuscript. If your ethics statement is written in any section besides the Methods, please delete it from any other section.

Response: Thank you for your explanation. Ethics statement has been modified in which only written in Methods section. All ethics statement written in other section other than as suggested section is deleted.

Reviewer #1: The authors analysed 67,944 records and report the prevalence of TB in different areas of Indonesia comparing sputum AFB, sputum culture, sputum genetic test, and chest radiograph. The authors use only the prevalence and the confidence interval to compare the different groups. The study reports the classification of three large islands in Indonesia: the island of Sumatra, the island of Java-Bali, and other islands (eastern part of Indonesia) with a classification of urban and rural areas.

Response: Thank you for your time in reviewing the manuscript.

Reviewer #1: Although the title is attractive, they do not conduct any geographic study.

Response: Thank you for the advice so we already changed the term to regional variation to minimize confusion.

Reviewer #1: The authors did not discuss the prevalence of TB in other developing countries, which would contribute to the conceptualization of TB at the global level.

Response: Thank you very much for improving the generalizability of our work. The discussion about contribution to the conceptualization of TB at the global level by using TB prevalence in developing countries is provided in the Discussion section as advised (Line 213-220, 225-231). As this study can be beneficial for other developing countries that has a wide geographics boundaries especially the country that has many islands, we specifically included the Philippines as one of the reference countries, because the Philippines has the similar island patterns with Indonesia.

Reviewer #1: Although they conclude that diagnostic approaches confuse tuberculosis, they do not mention full genomic sequencing as an alternative.

Response: Thank you very much for pointing that out. We have addressed this suggestion as a study limitation in the Discussion section. 

Reviewer #1: Then it would be convenient to modify the discussion by including data from other countries.

Response: Thank you for your suggestion. The data from other countries added as suggested.

Reviewer #2: This is an interesting study represents an analysis of Variation of TB prevalence across diagnostic approaches and 5 geographical areas of Indonesia. This article could be published in the Journal PLOS ONE after a major revision. I have some comments on it.

Response: Thank you very much for your time in reviewing the manuscript. We are pleased to know that our study is interesting, and your comments are very constructive and useful to develop it. 

Reviewer #2: Introduction: In line 53 “MTb” name must be in italics; review of the rest of the document.

Response: The term Mycobacterium tuberculosis in the manuscript was italicized. 

Reviewer #2: Methodology: The methodology does not describe the diagnostic tests used and the quality standards used to validate the results. Explain more about the methods used.

Response: Thank you very much for the comment. We did not collect the detail specification of each diagnostic test at each institution; however, the description of each diagnostic test was expanded by using information provided by the Indonesian Ministry of Health.

Reviewer #2: Results: The authors must show the risk factors of the patients (HIV, DM, malnutrition, immigration, etc.) and their association with the results in the diagnostic tests, in a table.

Response: Thank you for pointing out, unfortunately we did not have individual-level data on comorbidities. Nevertheless, we added one more table that shows the cases of a relevant comorbidities across regions in Indonesia based on a publish literature (Table 1).

Reviewer #2: What is the proportion of negative and positive culture in the different geographical areas? And how much is the culture, contributing to the timely diagnosis in these cases?.

Response: Thank you very much for pointing that out. We have addressed this suggestion as a study limitation in the Discussion section.

Reviewer #2: How do the authors explain that the prevalence of the positive culture is higher in the eastern region of Indonesia, compared to the results of the Chest X-ray and sputum AFB methods where the positive results are higher for others regions?.

Response: Thank you for the comments. In the eastern region of Indonesia, there are more sub-regions than other regions that are still underdeveloped and undeveloped; the education level is relatively low and the number of uneducated people in understanding tuberculosis is high. That effect sputum smear test might not be done properly and correctly at home. In addition to that, limited number of health workers who can assist in performing repeated sputum smear sampling and lack of medical facilities such as a TB detection device in the form of an up-to-date chest x-ray machine can affect the test results. Another possible interpretation of a negative smear test result is that the number of Mycobacterium tuberculosis bacteria is too small so that it cannot be detected through a microscope and further tests are needed by performing a microscopic examination of sputum or culture.

Reviewer #2: The authors could include data on the prevalence of migration-related TB cases, do you have any data on tuberculosis in migrants that you can include?.

Response: Thank you for the advice. We are sorry that our dataset does not include migrants or migration-related TB cases so we could not calculate the prevalence of migration-related TB. However, we added the Indonesian migrant data based on other literatures.

Reviewer #2: In your study you didn’t differentiate between active and latent TB, or the observed results include both?.

Response: In our study, the observed results included both active and latent TB.

Reviewer #2: The discussion part is very short, need to discuss more issues of “person’ place, and time”.

Response: Thank you so much for your comments, in fact your points above already help us expand the discussion be much more comprehensive and we are very thankful for that. We also added some more discussion on the time.

Reviewer #2: In the conclusion, could the authors explain from their point of view, some reasons for the spread of TB in the different geographical areas of Indonesia and their strategies or suggestions to improve early detection?.

Response: Additional point of view about reason for regional distribution of TB and strategies to improve early TB detection was provided in the Conclusion section. Education equality, rural and underdeveloped areas, low number of health worker, lack of health facilities and medical equipment were the major determinants for TB distribution along regional areas in Indonesia. We suggested several strategies to enhance TB initial detection: the government need to allocate extra budget for adding medical equipment which distributed evenly across geographical areas and allow public access to standardize laboratories especially for chest X-rays. Chest X-rays play a significant role to detect TB symptoms and was considered as the gold standard for initial TB confirmation in Indonesia.

Reviewer #2: The article has some mistakes in English language and have to be corrected.

Response: Thank you very much. We did a proof-reading service prior to resubmit the manuscript to review and enhance in English language. 

We hope that our responses are satisfactory. Should there be anything that might improve our work, please kindly inform us. Thank you very much for your kind consideration.

Best Regards,

Assoc. Prof. Dr. Krit Pongpirul, MD, MPH, PhD.

On behalf of the authors

---

## [Decision Letter · Decision Letter 1]

6 Oct 2021

Variation of tuberculosis prevalence across diagnostic approaches and geographical areas of Indonesia.

PONE-D-21-15288R1

Dear Dr. Pongpirul,

We’re pleased to inform you that your manuscript has been judged scientifically suitable for publication and will be formally accepted for publication once it meets all outstanding technical requirements.

Kind regards,

Frederick Quinn

Academic Editor

PLOS ONE

Additional Editor Comments (optional):

Reviewers' comments:

Reviewer's Responses to Questions

**Comments to the Author**

1. If the authors have adequately addressed your comments raised in a previous round of review and you feel that this manuscript is now acceptable for publication, you may indicate that here to bypass the “Comments to the Author” section, enter your conflict of interest statement in the “Confidential to Editor” section, and submit your "Accept" recommendation.

Reviewer #1: All comments have been addressed

Reviewer #2: All comments have been addressed

2. Is the manuscript technically sound, and do the data support the conclusions?

Reviewer #1: Yes

Reviewer #2: Yes

3. Has the statistical analysis been performed appropriately and rigorously? 

Reviewer #1: Yes

Reviewer #2: Yes

4. Have the authors made all data underlying the findings in their manuscript fully available?

Reviewer #1: Yes

Reviewer #2: Yes

5. Is the manuscript presented in an intelligible fashion and written in standard English?

Reviewer #1: Yes

Reviewer #2: Yes

6. Review Comments to the Author

Reviewer #1: The authors have addressed all observations made to the original manuscript. There are no more observations to the article. Hopefully, the study will have a good impact on the area.

Reviewer #2: The comments were satisfactorily resolved by the authors, so I consider that the study entitled "Variation of tuberculosis prevalence across diagnostic approaches and geographical areas of Indonesia" could be accepted for publication by the PLOSONE Journal. Yours sincerely PhD. Armando Martinez-Guarneros.

7. PLOS authors have the option to publish the peer review history of their article (what does this mean?). If published, this will include your full peer review and any attached files.

Reviewer #1: No

Reviewer #2: **Yes: **José Armando Martinez-Guarneros

---

## [Editor Report · Acceptance letter]

8 Oct 2021

PONE-D-21-15288R1 

Variation of tuberculosis prevalence across diagnostic approaches and geographical areas of Indonesia. 

Dear Dr. Pongpirul:

I'm pleased to inform you that your manuscript has been deemed suitable for publication in PLOS ONE. Congratulations! Your manuscript is now with our production department. 

Kind regards, 

on behalf of

Dr. Frederick Quinn 

Academic Editor

PLOS ONE